# Pathogenicity of *Purpureocillium lilacinum* and *Clonostachys rosea* against fall armyworm (*Spodoptera frugiperda*) under laboratory conditions

**Abel Jonathan Mussa**[1]*, **Sija Kabota**[1,2], **Joseph O Ruboha**[1,3], **Martin John Martin**[4], **Maulid W Mwatawala**[1]

**1** Department of Crop Science and Horticulture, Sokoine University of agriculture, Morogoro, Tanzania,
**2** Research, Consultancy and Publication Unit, National Sugar Institute (NSI), Kidatu-Morogoro, Tanzania,
**3** Department of Agricultural Sciences, Sokoine University of agriculture, Mpanda-Katavi, Tanzania,
**4** Institute of Pest Management IPM, Sokoine University of agriculture, Morogoro, Tanzania

* abel.mussa@sua.ac.tz

## Abstract

### Background

Fall armyworm, *Spodoptera frugiperda* (J.E. Smith) threatens staple crops across Africa. Integrating entomopathogenic fungi into Integrated Pest Management (IPM) offers a sustainable alternative to sole reliance on insecticides. This study quantified the pathogenicity of *Purpureocillium lilacinum* and *Clonostachys rosea* against *S. frugiperda* under controlled conditions.

### Methods

Second-fifth instar larvae and eggs were exposed to $1 \times 10^7$, $1 \times 10^8$, and $1 \times 10^9$ conidia mL$^{-1}$ of each fungus; sterile water served as control. Mortality was recorded over 3–9 days after treatment (DAT); feeding reduction was measured gravimetrically. Larval mortality was analyzed with GLMs/GLMMs (binomial-probit); feeding reduction by ANOVA/Tukey; LD50 and LT50 were estimated from dose-response models.

### Results

Larval mortality was significantly affected by concentration × time interaction and declined with advancing larval stage. Peak larval mortality was reached at a concentration of $1 \times 10^9$ conidia mL$^{-1}$ at 9 DAT. Feeding consumption reduction was significantly affected by larval instar, EPF species, and instar × concentration. Feeding reduction reached 60−74% in early instars at the highest dose. Egg mortality was primarily concentration-dependent with maximum values (up to 82% and 88% for *P. lilacinum* and *C. rosea*, respectively) at the dose of $1 \times 10^9$ conidia mL$^{-1}$ highest dose. Our findings supported the study hypothesis that efficacy of entomopathogenic fungi

**Data availability statement:** All data supporting the findings of this study are openly available. All datasets and analysis scripts supporting our study have been uploaded to the Zenodo repository and are publicly accessible at DOI: 10.5281/zenodo.18264488. Any additional materials are provided in the Supporting Information files.

**Funding:** This research was supported by the Sokoine University of Agriculture internal research fund. The funders had no role in study design, data collection/analysis, decision to publish, or preparation of the manuscript.

**Competing interests:** The authors declare that there is no conflict of interests regarding the publication of this Manuscript.

against *S. frugiperda* is primarily driven by interaction of spore concentration and exposure time across the host developmental stages, rather than the interaction of fungal species. The consistent susceptibility of early instars and the strong concentration-dependent responses highlight the functional potential of these native fungi as biologically relevant components of sustainable IPM strategies.

## Conclusions

Native *P. lilacinum* and *C. rosea* display dose-, stage-, and time-dependent pathogenicity and feeding suppression against *S. frugiperda*. These species are promising candidates for IPM; field validation and formulation optimization are the next steps.

---

## Introduction

Fall armyworm (*Spodoptera frugiperda* J.E. Smith) is one of the most destructive invasive pests threatening food security in Sub-Saharan Africa. Native to the tropical and subtropical regions of the Americas, *S. frugiperda* was first detected in Africa in 2016 [1] and subsequently reported in Tanzania in 2017 [2,3]. Since its invasion, the pest has spread rapidly across multiple agroecological zones, infesting major crops such as maize, sorghum, rice, and legumes. Yield losses attributed to *S. frugiperda* are severe and vary by country and production system. In Kenya, for example, maize yield losses range from 32–47%, while in Ethiopia they average around 32% [4]. These reductions are equivalent to thousands of tons of maize annually, and because maize is the primary staple crop in much of Sub-Saharan Africa, the implications for food security are profound. At the regional scale, economic losses have been estimated to exceed US$13 billion annually [2,5]. The destructive capacity of *S. frugiperda* stems from its larval feeding habit, in which caterpillars aggressively consume foliage, whorls, and reproductive structures, leading to defoliation, stalk breakage, and poor grain filling. In heavily infested fields, infestation levels can reach 100%, resulting in near-total crop failure [6]

In Tanzania, management of *S. frugiperda* has primarily relied on the application of synthetic insecticides, valued for their rapid action and availability [3,5,7]. Nonetheless, continuous reliance on chemical control has raised critical concerns. Resistance to active ingredients has already been documented in several countries, reflecting the pest's genetic adaptability and high reproductive potential. Moreover, synthetic insecticides pose risks to beneficial arthropods, pollinators, and natural enemies, thereby disrupting ecological balance and resilience of farming systems [8–10]. Environmental pollution, human health hazards, and escalating input costs further constrain the sustainability of insecticide-based strategies. These challenges underscore the urgent need for ecologically sound and locally adaptable alternatives to chemical control, particularly the development of biological control agents that are compatible with integrated pest management (IPM).

Entomopathogenic fungi (EPFs) have emerged as an important component of biological pest management. These fungi are naturally occurring insect pathogens with the

ability to infect and kill more than 700 insect species across different orders [11]. Unlike other microbial control agents that often require ingestion, EPFs are unique in their mode of action: they penetrate directly through the insect cuticle via enzymatic activity, colonize the hemocoel, and proliferate within the host, ultimately causing death through nutrient depletion and toxin production [12–15]. This capacity enables them to infect foliar as well as soil-dwelling pests, providing a wide ecological reach. Several genera including *Beauveria*, *Metarhizium*, *Isaria* (syn. *Cordyceps*), and *Purpureocillium* have been extensively studied, leading to the development of commercial mycoinsecticides [16,17]. Their ecological safety, compatibility with other IPM components, and potential to reduce pesticide dependence make them central to sustainable and agroecological pest management strategies.

Recent advances in entomopathogenic fungi research have reinforced their promise as biological control tools. For instance, Shehzad et al. [18] emphasized the ecological safety of EPFs as natural biocontrol agents, while Vivekanandhan et al. [19], and Gielen et al. [20] highlighted EPFs as effective and sustainable options, documenting continued innovation in formulation, application methods, and strain improvement. Both laboratory and field studies have repeatedly confirmed their insecticidal efficacy against major agricultural pests. *Beauveria* and *Metarhizium* species, in particular, have demonstrated consistent performance against lepidopteran and coleopteran pests, including *S. frugiperda*, across diverse regions [21–28]. Alongside these well-established fungi, new candidates are gaining attention. *Purpureocillium lilacinum* (Thom) and *Clonostachys rosea* (Link) are emerging as multifunctional fungi with dual potential in agriculture. Traditionally recognized for their antagonism against soilborne plant pathogens, they are increasingly acknowledged for their capacity to infect and suppress insect pests [29–33]. Laboratory trials in Pakistan with *P. lilacinum* demonstrated significant insecticidal activity [34–37], while multiple studies have reported insecticidal efficacy of *C. rosea* against a wide range of pests [31,38–40].

Both *P. lilacinum* and *C. rosea* are cosmopolitan species, naturally occurring in soils, decaying organic matter, nematodes, and insect cadavers [32,41,42]. Beyond their insecticidal action, they contribute to plant growth promotion and biofertilization by enhancing nutrient uptake and suppressing plant pathogens [29,43,44]. These multifunctional attributes make them attractive candidates for integration into pest and soil health management programs. However, despite their global distribution, research on these fungi in Tanzania remains underdeveloped. While both species have been reported in the country, most studies focus on imported fungal formulations rather than exploring the diversity and efficacy of native isolates. Moreover, some locally available fungi have been found pathogenic to crops and remain unvalidated for their potential mycotoxin production, raising safety concerns [9,10,45]. The systematic isolation, characterization, and pathogenicity testing of native fungal strains against *S. frugiperda* and other insect pests have not yet been undertaken at scale, leaving a major gap in the development of homegrown biocontrol solutions.

No published study in Tanzania has yet evaluated the pathogenicity or virulence of native isolates of *P. lilacinum* and *C. rosea* against *S. frugiperda*. Given the pest's economic importance and the limitations of current insecticide-based control strategies, exploring the pathogenic potential of these fungi is both timely and necessary. Establishing the efficacy of native isolates could expand the repertoire of EPFs available for local pest management, reduce dependence on imported products, and support the development of context-specific biological control strategies.

This study evaluated the pathogenicity and virulence of native *P. lilacinum* and *C. rosea* isolates against multiple stages of *S. frugiperda* under laboratory conditions, focusing on larval and egg mortality as well as feeding reduction at different spore concentrations and exposure times. The findings broaden the diversity of EPFs available for pest control in Tanzania, clarify stage-specific susceptibility critical for management design, and provide baseline data for optimizing EPF-based interventions within IPM frameworks. By diversifying control options, the study supports reduced reliance on insecticides, improved resistance management, and more sustainable agriculture.

It was hypothesized that native isolates of *P. lilacinum* and *C. rosea* would exhibit dose, stage, and time-dependent pathogenicity against *S. frugiperda*. Specifically, younger larval instars were expected to show higher susceptibility due to weaker cuticular defenses and less developed detoxification mechanisms, while older larvae were predicted to be more resilient. Furthermore, it was anticipated that higher spore concentrations and longer post-treatment periods would result in greater mortality and stronger suppression of feeding activity.

## Materials and methods

### Study area

The study was conducted from August 2024 to February 2025 in the Mycological and Molecular Laboratories, Department of Crop Science and Horticulture, Sokoine University of Agriculture (SUA), Morogoro, Tanzania (6.8520°S, 37.6576°E). Morogoro Municipality is located on the lower slopes of the Uluguru mountains at an elevation of approximately 500–600 m above sea level. The area experiences a tropical sub-humid climate with mean annual rainfall ranging from 800 to 1,200 mm, distributed in two rainy seasons: the short rains from October to December and the long rains from March to May. Average annual temperatures range between 18°C and 30°C, with relatively cooler conditions during the rainy season and warmer conditions during the dry season [46]. The area is characterized by steep slopes, deeply dissected valleys, and diverse vegetation ranging from lowland woodlands to montane and cloud forests with diverse agricultural systems [47].

### Collection and rearing of test insect

*Spodoptera frugiperda* larvae were collected from maize fields using culture vials lined with cotton wool and reared following protocol of Idrees et al. [22] with minor modification. Preferably, fourth to sixth instar larvae were selected and transported to the laboratory for rearing. In the laboratory, larvae were transferred into sterile, aerated plastic containers and provided with fresh, sterile maize leaves. The leaves were replaced every 24 hours to maintain hygiene and ensure adequate nutrition. Once pupation occurred, the containers were supplemented with sterilized sand to facilitate pupal development. Pupae were then transferred to aerated rearing cages to allow for adult emergence.

Emerging adult moths were maintained in cages and supplied with a 10% honey solution soaked in cotton wool for sustenance. Potted maize plants were introduced into the cages to serve as oviposition substrates. Upon hatching, larvae were collected and transferred to new sterile plastic containers, provided with fresh maize leaves, and maintained under controlled laboratory conditions: 25±2°C temperature, 65±5% relative humidity, and a 12:12 h light: dark photoperiod. The colony was maintained for at least two successive generations to obtain sufficient numbers of target-stage larvae for bioassays [22]. Only second to fifth instar larvae were used for bioassay tests because first instars are extremely fragile and prone to mortality due to mishandling. On the other hand, sixth instars are less susceptible to EPF due to a thicker cuticle and could pupate during the assay, potentially confounding mortality assessments; thus, the selected instars provided an optimal balance between susceptibility and experimental reliability [48].

### Preparation of fungal suspension for bioassay

Entomopathogenic fungal species were obtained from soil samples collected along the slopes of the Uluguru Mountains through selective media isolation following the established protocol by Humber [49] and screened for pathogenicity. The pathogenic isolates were confirmed using morphological characters and ITS rDNA sequencing, which were compared against GenBank references. Single-spore of pathogenic isolates were cultured on Potato Dextrose Agar (PDA) and incubated at 25°C in darkness for two weeks. Conidia were harvested in sterile double-distilled water, filtered through sterile gauze to remove mycelial debris, and gently mixed to ensure a uniform suspension. Spore concentration was determined using a hemocytometer, and viability was confirmed before use. A stock suspension of $1 \times 10^8$ conidia mL$^{-1}$ was prepared and diluted or concentrated to $1 \times 10^7$, $1 \times 10^8$, and $1 \times 10^9$ conidia mL$^{-1}$ for bioassays [48].

### Bioassay of fungal species against immature stages of *S. frugiperda*

A pathogenicity test bioassay was conducted using a Completely Randomized Design (CRD) with three replicates per treatment to evaluate the efficacy of EPF species against immature stages of *S. frugiperda*. Each group of 10 larvae (second to fifth instar) and 20 eggs attached to maize leaves were separately placed in sterile, aerated plastic container

   

as described by [48] and [50]. Each group was uniformly sprayed once with 10 mL of each fungal suspension at concentrations of $1.0 \times 10^7$, $1.0 \times 10^8$, and $1.0 \times 10^9$ conidia mL$^{-1}$ using handheld sprayer, while a negative control received sterile distilled water. Sterile paper towels were placed beneath maize leaves in each container to absorb excess spray solution and removed thereof after 24 hours, and fresh, sterile maize leaves were provided daily under replacement as a food source to larvae. All test units were maintained at $25 \pm 2°C$, $65 \pm 5\%$ relative humidity, and a 12:12 h light: dark photoperiod throughout the observation period. Larval mortality was recorded every 24 hours for nine days post-inoculation, and egg mortality was determined by counting hatched and unhatched eggs nine days after treatment. Dead insects were removed, isolated in sterile conditions, and examined for typical signs of mycosis induced by EPFs.

## Assessment of feeding consumption reduction of *S. frugiperda* larvae

Feeding activity of *S. frugiperda* larvae was evaluated following the methodology of Idrees [22], with minor modifications to suit experimental conditions. Each group of 10 larvae from each of the 2nd, 3rd, 4th, and 5th instars were separately provided with 7 grams of fresh, sterile, and untreated maize leaves daily. To maintain consistency and prevent leaf desiccation or contamination, the maize leaves were replaced every 24 hours.

Feeding activity was quantified by measuring the weight of maize leaves before and immediately after the 24-hour feeding period. The difference in weight represented the amount of leaf material consumed by the larvae, allowing for an indirect assessment of the physiological effects of EPF treatments on larval feeding behavior. This approach effectively monitored larval consumption rates and served as a crucial parameter for evaluating the biocontrol potential of the fungal species.

To determine the impact of EPF treatments on feeding, the percentage reduction in feeding consumption was calculated using the formula:

$$\text{Feeding Consumption Reduction (\%)} = [(W_c - W_t)/W_c] \times 100$$

where:

$W_c$ = weight of leaf material consumed by the control (untreated) larvae
$W_t$ = weight of leaf material consumed by the treated larvae

## Ethics statement

This study protocol was reviewed and approved by the Ethics Review Board of the College of Agriculture, Sokoine University of Agriculture (SUA), Tanzania (Approval Number: SUA/DPRTC/MCS/D/2022/0021/07). Written informed consent to access and conduct research at the field sites and laboratory was waived by the Ethics Review Board of Sokoine University of Agriculture, which owns and manages all land and facilities where the study was carried out. As the study was conducted entirely within university-owned properties, no additional permits from external authorities were required. This study did not involve human participants, animals or human participants' data; therefore, written informed consent was not required.

## Statistical data analysis

All data on mortality and feeding reduction were analyzed using R version 4.4.3 [51]. Normality of feeding reduction data was assessed using the Shapiro-Wilk test. When assumptions were met, analysis of variance (ANOVA) followed by Tukey's HSD post hoc test ($\alpha = 0.05$) was used.

Larval mortality data, expressed as proportions, were analyzed using generalized linear models (GLMs) with a binomial error distribution and probit link function. To account for repeated measures and fungal species variation, generalized linear mixed-effects models (GLMMs) were fitted with conidial concentration and time after treatment as fixed effects and replicates nested within species as a random effect.

Lethal dose (LD$_{50}$) and lethal time estimates were obtained using the dose.p function from the MASS package. Model significance was evaluated with Wald z-tests and Type III $\chi^2$ tests using the car::Anova function. Model fit and residual diagnostics were assessed using the DHARMa package.

## Results

### Effect of *P. lilacinum* and *C. rosea* on larval mortality of *S. frugiperda*

We observed a significant effect of concentration × time interaction on larval mortality ($p = 0.01$; Table 1). This indicates that the effect of spore concentration on larval mortality was significantly dependent on the duration of exposure. Increasing spore concentration from 0 to $1 \times 10^9$ conidia mL$^{-1}$ caused a significant rise in mortality, with mortality reaching peak at 9 DAT. Post hoc analysis (Tukey HSD) showed that mortality at 9 DAT was significantly higher than that recorded at 3 and 6 DAT (Fig 1). Overall, mortality was significantly higher at 9 DAT across all concentrations and lowest at 3 DAT ($p < 0.001$).

Mortality also significantly varied with larval instar stage ($p < 0.001$), with second and third instars being more susceptible than fourth and fifth instars, irrespective of concentration and time (Fig 2). The main effect of fungal species was not significant ($p > 0.05$), although *C. rosea* caused slightly higher mortality (48.5% at 9 DAT) compared to *P. lilacinum* (43.7%).

Probit analysis showed that each additional day post-treatment increased mortality by 3−4%, while a tenfold increase in dose raised mortality by approximately 7% (S1Table). Conidial concentration and exposure time significantly affected mortality, with minimal random variation. The lethal dose (LD$_{50}$) was estimated at $1.6 \times 10^{12}$ conidia mL$^{-1}$, which representing the average estimated virulence across all tested instars. The median lethal time (P$_{50}$) decreased from 15.4 to 11.8 days with increasing dose, indicating clear dose- and time-dependent virulence. Dose-response curves further showed that second and third instars required lower conidial concentrations to reach LD$_{50}$ than fourth and fifth instars (Fig 3).

**Table 1. Analysis of variance (ANOVA) results for the effects of spore concentration, larval instar stage, and time after treatment, along with their interactions on *S. frugiperda* larval mortality caused by EPFs.**

| Source of variation | $\chi^2$ | Df | p-value |
|---|---|---|---|
| Concentration (C) | 70.1107 | 13 | *** |
| Instar Stage (I) | 52.11186 | 13 | *** |
| Days After Treatment (DAT) | 66.65455 | 8 | *** |
| Species (S) | 7.836619 | 5 | ns |
| C: S | 1.991494 | 7 | ns |
| **C: DAT** | 15.90452 | 10 | * |
| C: I | 4.006428 | 16 | ns |
| S: DAT | 1.53809 | 4 | ns |
| S: I | 1.915249 | 7 | ns |
| DAT: I | 2.711469 | 11 | ns |
| C: S: DAT | 0.763631 | 7 | ns |
| C: DAT: I | 2.132113 | 19 | ns |
| S: DAT: I | 1.931265 | 7 | ns |
| C: S: I | 1.166897 | 10 | ns |
| C: S: DAT: I | 1.248728 | 18 | ns |

Notes: Bold text indicates significant interaction effects. Asterisks indicate the level of significant difference: *** $p < 0.001$, ** $p < 0.01$, * $p < 0.05$, ns $p \geq 0.05$.

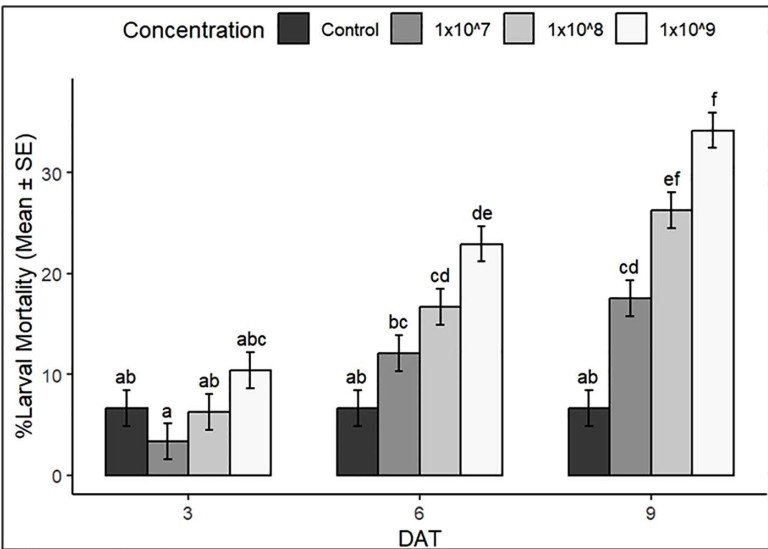

**Fig 1. Effect of conidial concentration and Days After Treatment (DAT) on mortality of *S. frugiperda* larvae.** Different letters above bars indicate significant differences among concentrations at each DAT (Tukey's HSD, p < 0.05). Mortality also increased significantly with DAT (p < 0.001), reflecting the time-dependent virulence of the entomopathogenic fungi. Error bars represent standard error of the mean (SEM).

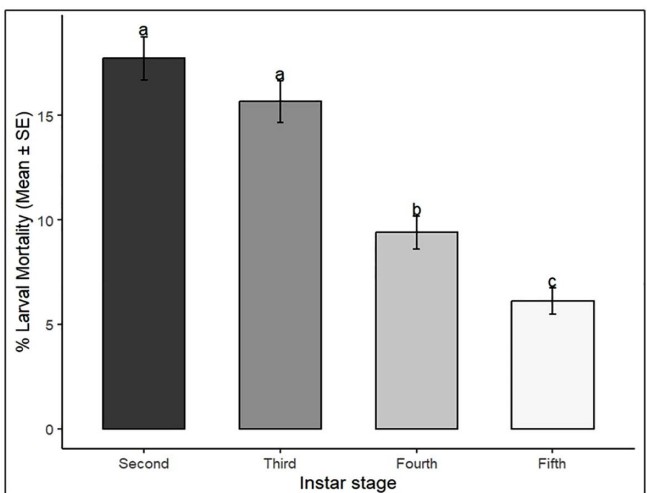

**Fig 2. Larval mortality of *S. frugiperda* across instar stages following treatment with *C. rosea* and *P. lilacinum*.** Bars show mean ± SE. Different letters above bars indicate significant differences among instars (Tukey's HSD, *p* < 0.05).

### Effects of *P. lilacinum* and *C. rosea* on feeding consumption reduction of *S. frugiperda* larvae

Feeding consumption reduction was strongly influenced by the interaction between concentration and larval instar stage (concentration x instar stage, *p* < 0.001), indicating that the effect of concentration on feeding depends on the developmental stage of the larvae. In contrast, fungal species-related interaction effects were not significant (*p* > 0.3). The main effects of spore concentration, fungal species and instar stage were also significant (all *p* < 0.001; Table 2). A clear concentration-instar dependent effect was observed (Fig 4), at $1 \times 10^9$ conidia mL$^{-1}$, *C. rosea* caused the highest reduction (74% in second

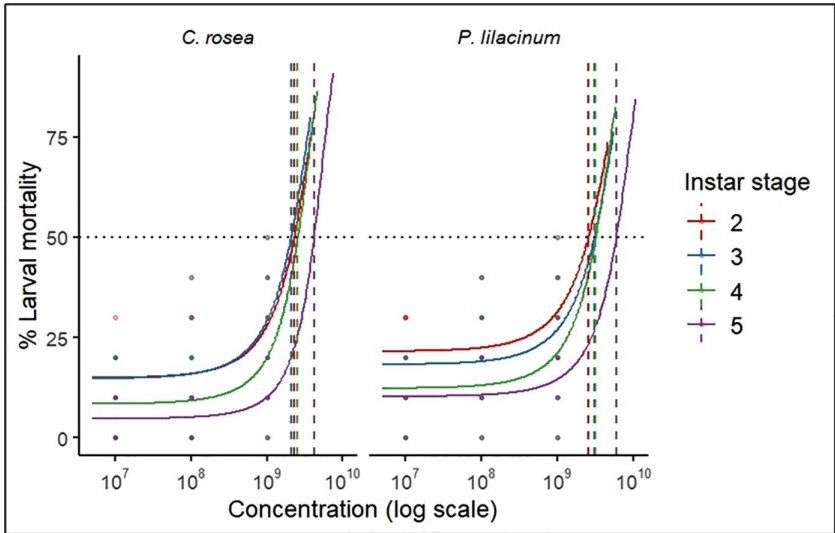

**Fig 3. Dose-response curves of *S. frugiperda* larval mortality across instar stages following treatment with EPFs.** Curves fitted with probit regression.

**Table 2. ANOVA summary table of the effects of different spore concentration of EPF species on feeding consumption reduction in different instars of *S. frugiperda*.**

| Source | Df | F-value | p-value |
|---|---|---|---|
| Concentration (C) | 2 | 999.94 | *** |
| Instar stage (I) | 3 | 1661.39 | *** |
| Species (S) | 1 | 28.03 | *** |
| C×I | 6 | 30.45 | ** |
| C×S | 2 | 0.16 | ns |
| I×S | 3 | 0.04 | ns |
| C×I×S | 6 | 6.55 | ns |

Notes: ANOVA performed on arcsine-transformed proportions. Asterisks indicate the level of significant difference: *** $p<0.001$, ** $p<0.01$, * $p<0.05$, ns $p \geq 0.05$.

instars, 66% in third instars). *P. lilacinum* achieved maximum reductions of 66% (second instars) and 60% (third instars) at the same dose. Both fungi were less effective against fourth and fifth instars (<45% and <35%, respectively).

## Effects of *P. lilacinum* and *C. rosea* on egg mortality

ANOVA showed that spore concentration had a highly significant effect on egg mortality ($p<0.001$), with mortality increasing at higher concentrations of both fungal species (Table 3). Neither the main effect of fungal species ($p=0.2291$) nor its interaction with concentration was significant ($p=0.9707$), indicating that egg mortality was primarily determined by spore concentration, rather than fungal species.

At $1 \times 10^9$ conidia mL$^{-1}$, *C. rosea* caused maximum egg mortality of 88.3%, while *P. lilacinum* achieved 81.7%. Mean egg mortality at $1 \times 10^9$ conidia mL$^{-1}$ was significantly higher than at lower concentrations and the control, as indicated by Tukey HSD grouping letters (Fig 5).

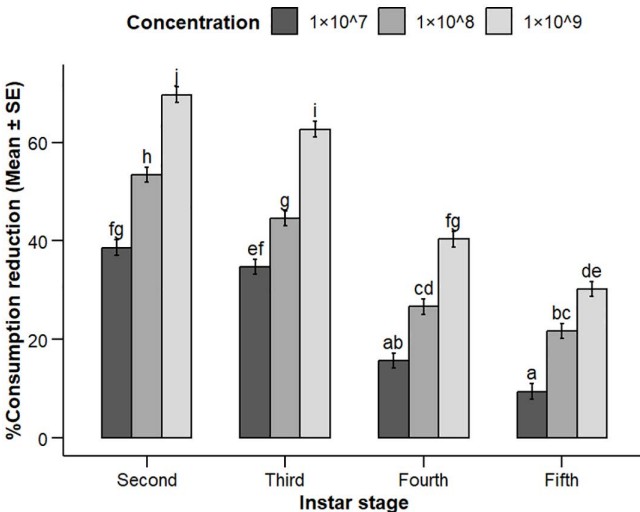

**Fig 4. Feeding reduction (%) of *S. frugiperda* larvae treated with EPFs at three concentrations (1 × 10⁷, 1 × 10⁸, and 1 × 10⁹ conidia mL⁻¹). Bars show mean ± SE. Different letters indicate significant differences (*p* < 0.05, Tukey's HSD).**

**Table 3. ANOVA summary table of the effects of different spore concentration of EPF species on eggs mortality of *S. frugiperda*.**

| Source of Variation | F-value | Df | p-value |
|---|---|---|---|
| Concentration | 186.47894 | 3 | *** |
| Species | 1.4463557 | 1 | ns |
| Concentration: Species | 0.2409136 | 3 | ns |

Notes: Asterisks indicate the level of significance: *** p < 0.001, ns = p ≥ 0.05.

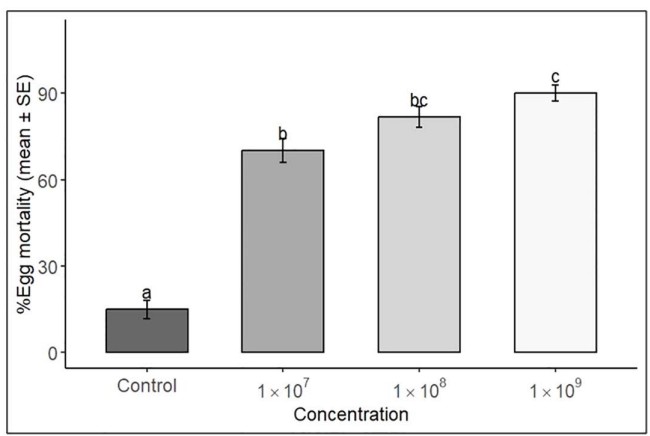

**Fig 5. Egg mortality of *S. frugiperda* in response to *C. rosea* and *P. lilacinum* treatments at different concentrations.** Bars show mean ± SE. Different letters indicate significant differences among concentrations (*p* < 0.05, Tukey's HSD).

## Discussion

Our results confirm that both *Purpureocillium lilacinum* and *Clonostachys rosea* possess measurable insecticidal potential against *S. frugiperda*, with performance varying by developmental stage, fungal species, and conidial concentration. As expected, larval mortality increased with higher inoculum levels, while early instars were more susceptible than older larvae, demonstrating a clear dose- and age-dependent virulence pattern. Similar findings have been reported for other entomopathogenic fungi. In Tanzania, *Metarhizium anisopliae*, *Fusarium* spp., and *Beauveria bassiana* caused greater mortality in early instars of *S. frugiperda*, with effects enhanced by higher doses and longer exposure times [9,10,52]. Comparable stage- and dose-dependent responses have also been documented in Kenya and Ethiopia, where *Metarhizium* spp. and *Beauveria* spp. produced significant larval mortality [23,26].

In our assays, *P. lilacinum* showed particularly high pathogenicity against eggs and neonates, with mortality exceeding 80% at $1 \times 10^9$ conidia mL$^{-1}$, but efficacy declined against later instars, often falling below 50% at the same concentration. This stage-specific dose-response pattern has been widely observed. Laboratory bioassays confirmed that *S. frugiperda* eggs and neonates are highly susceptible, with mortality above 95% at $10^6 - 10^8$ conidia mL$^{-1}$, while 1st-2nd instars required higher concentrations, yielding LD$_{50}$ values of $1.38 \times 10^8 - 2.56 \times 10^8$ conidia mL$^{-1}$ by day 7 [30,37]. In Egypt, Suzan et al. [53], similarly reported larval mortality of 37.5% to 68.5% across a gradient of $10^5 - 10^9$ conidia mL$^{-1}$. Beyond *S. frugiperda*, *P. lilacinum* has demonstrated strong activity against other lepidopterans: *Galleria mellonella* larvae exhibited LD$_{50}$ values as low as $3.1 \times 10^4$ conidia mL$^{-1}$, with LT$_{50}$ around 1−2 days at $10^8$ conidia mL$^{-1}$ [54]. while high mortality has also been reported in *Tuta absoluta* and *Spodoptera litura* [35]. The fungus is also effective against insect orders such as Hemiptera and Diptera, although feeding larvae in these groups typically require higher doses [33,36,55–57]. Importantly, *P. lilacinum* produces sublethal effects even at lower inoculum levels, including delayed larval development, reduced pupal weight, and impaired adult emergence [58]. These additional impacts suggest that its influence extends beyond direct mortality.

The value of *P. lilacinum* is further enhanced by its activity against plant-parasitic nematodes, particularly *Meloidogyne* spp., where suppression is dose-dependent [59,60]. It is also capable of colonizing plants as an endophyte, thereby providing systemic protection against both insects and pathogens, while stimulating plant growth [58,59]. Its production of secondary metabolites, such as leucinostatins and paecilotoxins, likely underpins both insecticidal activity and feeding inhibition [55,61]. Together, these attributes position *P. lilacinum* as a versatile biocontrol agent that is most effective when applied early in pest infestations, with strong potential for integration into diversified pest management programs.

Although direct studies on *C. rosea* against *S. frugiperda* are lacking, evidence from other pests highlights its significant insecticidal capacity. In lepidopterans such as *G. mellonella* and *T. absoluta*, *C. rosea* caused up to 97% mortality [31,62]. In stored-product beetles including *Callosobruchus maculatus*, *Trogoderma granarium*, and *Tribolium castaneum*, mortality rates of 71−76% were achieved at $10^8$ conidia mL$^{-1}$, with LT$_{50}$ values of 4.8–5.0 days [63]. Against hemipterans, moderate to high activity has been reported: *Diaphorina citri* showed up to 47% mortality at $10^8$ conidia mL$^{-1}$, while *Amritodus atkinsoni* reached nearly 97% mortality at $3 \times 10^8$ conidia mL$^{-1}$ after 7 days [39,40]. Efficacy has also been demonstrated across Coleoptera, Hemiptera, and Hymenoptera in other regions [38,64–68]. In Tanzania, *C. rosea* has been naturally isolated from *S. frugiperda* cadavers [41], confirming ecological association, though its direct virulence against this pest remains untested. Beyond insect pathogenicity, *C. rosea* plays a dual role as a mycoparasite, antagonizing plant pathogens such as *Botrytis cinerea* and *Fusarium* spp. in a dose-dependent manner [29,34,69], while also promoting plant growth through endophytic colonization [31,44]. Its mechanisms of action include direct infection of insect hosts, rhizosphere competition, and the production of antifungal and insecticidal metabolites [61], enhance its versatility in agroecological systems. Furthermore, its compatibility with IPM programs and its safety to beneficial insects support its suitability for sustainable agriculture [70]. However, compared with *P. lilacinum* and classical genera such as *Beauveria* and *Metarhizium*, insect-focused dose-response studies of *C. rosea* remain scarce, highlighting a key knowledge gap and an opportunity for future research.

Both fungi displayed strong stage-dependent efficacy, with eggs and neonates consistently more vulnerable than feeding larvae. $LD_{50}$ values for early developmental stages were often an order of magnitude lower than for later instars [30,53]. This pattern reflects fundamental differences in insect physiology: younger larvae possess softer cuticles, lower levels of detoxification enzymes, and weaker immune responses, whereas older instars develop thicker cuticles and more robust defenses that limit fungal penetration and proliferation [19,71,72]. Such age-dependent susceptibility has been consistently observed across other entomopathogenic fungi [50,73], reinforcing the importance of applying these fungi early in pest infestations to maximize biocontrol efficacy.

The performance of *P. lilacinum* and *C. rosea* should also be viewed in the broader context of entomopathogen research. Although *Beauveria* and *Metarhizium* remain the most widely studied and commercialized fungi [18], emerging evidence indicates that less-studied taxa can complement or, under certain circumstances, outperform them. Both *P. lilacinum* and *C. rosea* are ecologically relevant, occurring in diverse soils and frequently associated with insect cadavers. Their ability to induce multiple outcomes including direct mortality, feeding inhibition, reduced fecundity, and developmental delays further enhances their potential. Nonetheless, gaps remain in understanding their persistence, dispersal, and sensitivity to environmental factors such as UV radiation and humidity, which often constrain field performance [20].

In Africa, integrated biological control approaches are gaining attention. Ngangambe et al. [9], demonstrated enhanced suppression of *S. frugiperda* when entomopathogenic fungi were combined with parasitoids, compared with single-agent applications, suggesting valuable synergistic effects. Such findings highlight the potential for *P. lilacinum* and *C. rosea* to be deployed not only as stand-alone agents but also as components of multi-enemy strategies tailored to smallholder farming systems. Moreover, ecological approaches such as push-pull technology in Kenya have been highly effective against *S. frugiperda*. By intercropping maize with *Desmodium* (push) and planting Napier or Brachiaria grasses as trap crops (pull), farmers achieved significant reductions in infestation and increased yields [74]. Integrating entomopathogenic fungi into these systems could add another layer of control, as their ability to cause mortality, inhibit feeding, and reduce reproduction would complement the deterrent and trap functions of push-pull. This diversification of tactics reduces reliance on synthetic insecticides and lowers the risk of pest adaptation, thereby enhancing the resilience of maize-based agroecosystems.

Overall, both *P. lilacinum* and *C. rosea* demonstrate strong potential as biological control agents against *S. frugiperda*, with efficacy tightly linked to spore concentration and insect developmental stage. Eggs and neonates remain the most susceptible, with $LD_{50}$ values often an order of magnitude lower than later instars, and reliable control typically requiring $\geq 10^7$ conidia $mL^{-1}$ within 5–7 days. These results highlight the importance of early intervention and the consideration of multiple outcomes when evaluating fungal biocontrol agents.

## Conclusion

This study demonstrates that native isolates of *P. lilacinum* and *C. rosea* possess significant potential as biological control agents against *S. frugiperda* in Tanzania. Their efficacy was strongly influenced by the interaction between spore concentration and time after treatment (DAT), indicating that the impact of spore dose depends on exposure duration, with mortality increasing over time at higher concentrations. In addition, spore concentration, insect developmental stage, and DAT each exerted significant independent effects, with eggs and neonates consistently showing higher susceptibility than later instars. Beyond direct mortality, both fungi induced sublethal effects such as reduced feeding, which could further contribute to crop protection. These findings broaden the spectrum of entomopathogenic fungi available for integrated pest management (IPM) and highlight the value of exploiting locally adapted strains to reduce reliance on chemical insecticides. Future research should focus on validating these findings under African field conditions to confirm persistence and performance under variable agroecological conditions and to integrate molecular and biochemical tools to better understand host-pathogen interactions. Moreover, evaluating compatibility with other agroecological practices such as parasitoids, intercropping, and push-pull systems will be critical for designing resilient and farmer-friendly IPM strategies. By

advancing the use of indigenous fungal resources, this work provides a foundation for sustainable, environmentally safe, and context-specific management of fall armyworm in maize-based smallholder systems.

## Supporting information

**S1 Table. Probit analysis of *S. frugiperda* larval mortality in response to spore concentration and time after treatment of EPF*s*.** Values include estimates of lethal dose ($LD_{50}$), and median lethal time ($P_{50}$).
(DOCX)

**S1 Fig. Pathogenic fungal colonies and *Spodoptera frugiperda* larvae showing signs of mycosis caused by entomopathogenic fungi: (a) *C. rosea* colony (b) *P. lilacinum* colony (c) Larva infected with *C. rosea* at 3 DAT, showing exoskeleton shedding; (d) Dead larva treated with *C. rosea* at 5 DAT; and (e) dead larva treated with *P. lilacinum* at 7 DAT.**
(TIF)

## Acknowledgments

We thank God Almighty for guidance throughout this study. We acknowledge Sokoine University of Agriculture (SUA) for research funding and support, and our families for their encouragement and prayers. We acknowledge the use of AI tools (ChatGPT, OpenAI) to check gramma, check spelling, improve clarity and reduce redundancy in the text. All content was reviewed and approved by the authors, who take full responsibility for the final manuscript.

## Author contributions

**Conceptualization:** Abel Jonathan Mussa, Maulid W Mwatawala.

**Data curation:** Abel Jonathan Mussa.

**Formal analysis:** Abel Jonathan Mussa, Sija Kabota, Joseph O Ruboha.

**Funding acquisition:** Maulid W Mwatawala.

**Investigation:** Abel Jonathan Mussa.

**Methodology:** Abel Jonathan Mussa, Martin John Martin.

**Project administration:** Maulid W Mwatawala.

**Resources:** Martin John Martin.

**Supervision:** Martin John Martin, Maulid W Mwatawala.

**Writing – original draft:** Abel Jonathan Mussa.

**Writing – review & editing:** Sija Kabota, Joseph O Ruboha, Martin John Martin, Maulid W Mwatawala.

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
