## [Decision Letter · Decision Letter 0]

10 Dec 2025

Dear Dr. Abel Jonathan Mussa,

Thank you for submitting your manuscript to PLOS ONE. After careful consideration, we feel that it has merit but does not fully meet PLOS ONE’s publication criteria as it currently stands. Therefore, we invite you to submit a revised version of the manuscript that addresses the points raised during the review process.

We look forward to receiving your revised manuscript.

Kind regards,

Yêyinou Laura Estelle Loko

Academic Editor

PLOS One

Journal Requirements:

Additional Editor Comments:

Reviewer 1

Spodoptera frugiperda is an important pest for the EPPO region with more than a 1000 host species. Itt is a priority quarantine pest for the European Union, and quarantine pest for Egypt, Morocco and Tunisia on the African continent.

The manuscript (MS) conforms to the journal's requirements.

Introduction

It appears that at least parts of the introduction is AI generated according to copyleaks AI detector.

There should be a declaration at the end of the MS in the Acknowledgements as outlined be the journal here:https://journals.plos.org/plosone/s/ethical-publishing-practice#loc-artificial-intelligence-tools-and-technologies

Materials used

line 180: please cite or describe the selective media isolation

line 195: please describe how was the treatment carried outand how long were the larvae exposed to the spores of the different fungi, and when were the paper towels exchanged under the larvae.

Expression of the treatment in l/ha would be also useful for practicing pest managers, so they could compare the dosage to already existing products.

Results

line 267: I think that the supplementary table could be included after this sentence or paragraph, because it contains relevant and valuable information.

lines 279, 289, : "species related interactions" and "Neither species identity" would be more cleared if the authors would include the fungi or fungal words.

I would appreciate images of the fungal colonies the healthy and infected larvae and images of the eaten corn leaves from each treatment to make the MS more visually appealing.

Reviewer 2

My Specific Comments for Authors

Please address the following points to enhance the clarity and completeness of the manuscript. I believe no additional experiments are required.

1. Abstract and Conclusion Clarity

i. Repetitive Content: The final paragraph of the Abstract, lines 37–43, largely repeats the detailed findings summarized in the preceding Results paragraph, lines 30–37. To improve conciseness, consider synthesizing the final paragraph to focus purely on the validation of the hypothesis and the significance of the findings, rather than re-stating the quantitative results.

ii. Conclusion Sentence Refinement: In the Conclusions section (lines 408-424), the sentence "Their efficacy was strongly influenced by spore concentration and insect developmental stage, with eggs and neonates consistently showing higher susceptibility than later instars" is highly accurate but could be enriched by briefly mentioning the crucial third factor confirmed by your analysis. Table 1 shows that Time After Treatment (DAT) was also a highly significant factor in larval mortality (p < 0.001), with a significant concentration x time interaction (p = 0.010). Please ensure the conclusion reflects all three major factors demonstrated in the data (Dose, Stage, and Time) for complete accuracy.

2. Methods Documentation

i. Fungal Isolate Characterization: The Methods section states that the EPF species were obtained from soil samples collected in the Uluguru Mountains, isolated through selective media, and screened for pathogenicity. For full reproducibility, please briefly clarify how the specific identity of the native isolates (i.e., confirming they were truly Purpureocillium lilacinum and Clonostachys rosea) was achieved. Was this identification based solely on morphology, or was a confirmatory method (such as molecular sequencing/PCR) used? This clarification should be added around lines 178-187.

ii. Rearing Conditions for S. frugiperda: The rearing conditions are well-described (25±2°C, 65±5% relative humidity, 12:12 h L:D) relative humidity, 12:12 h L:D). Please confirm in the Materials and Methods (lines 198-199) that all bioassay test units (larvae and eggs treated with EPFs or control) were maintained under these identical controlled conditions throughout the 9-day observation period, as consistency in environmental parameters is critical for EPF bioassays.

3. Results and Discussion Context

i. LD50 Value Context: The Results section reports that the overall lethal dose (LD50) was estimated at 1.6 x 1012 conidia mL-1. While the manuscript correctly notes that second and third instars required lower concentrations to reach LD50 (Fig 3), the single combined LD50 value of 1.6 x 1012 is significantly higher than most reported values for EPFs against early-stage lepidopterans (e.g., 106 - 108 conidia mL-1 cited in the Discussion for similar assays). Please confirm in the text (around lines 268–269) that this high calculated LD50 (1.6 x 1012) represents the average estimated virulence across all tested instars (L2 to L5), which includes the significantly less susceptible late instars, thus explaining the high magnitude. This clarification will prevent potential misinterpretation of the strain’s overall efficacy.

ii. Justification for Instar Selection: The rationale for excluding the 1st and 6th instars is provided in lines 202-207. Please move this detailed justification to the end of the Collection and rearing of test insect subsection (around line 176), or within the Bioassay subsection, rather than placing it after the description of the mortality measurement protocols (lines 199-202). This slight reorganization will improve the flow of the Methods section.

4. Figure and Table Presentation

i. Figure 1 Legend Clarity: Figure 1 shows the effect of concentration and time on mortality. Table 1 shows that both Concentration and DAT were highly significant. The legend for Fig 1 states: "Different letters above bars indicate significant differences among concentrations (Tukey’s HSD, p < 0.05)". Please ensure the figure itself (or its associated legend/notes) also clearly communicates the significance found across the Days After Treatment (DAT), as this is a fundamental and highly significant finding (p < 0.001).

Provided images of the healthy and infected larvae and the feeding damage from the treatments.

This is a valuable contribution to the literature on biological control in Sub-Saharan Africa. The minor revisions requested aim only to strengthen the presentation and ensure that the powerful statistical conclusions are fully supported by clear text and methodology.

Reviewers' comments:

Reviewer's Responses to Questions

**Comments to the Author**

1. Is the manuscript technically sound, and do the data support the conclusions?

Reviewer #1: Yes

Reviewer #2: Yes

2. Has the statistical analysis been performed appropriately and rigorously?

Reviewer #1: Yes

Reviewer #2: Yes

3. Have the authors made all data underlying the findings in their manuscript fully available?

Reviewer #1: Yes

Reviewer #2: Yes

4. Is the manuscript presented in an intelligible fashion and written in standard English?

Reviewer #1: Yes

Reviewer #2: Yes

Reviewer #1: Spodoptera frugiperda is an important pest for the EPPO region with more than a 1000 host species. Itt is a priority quarantine pest for the European Union, and quarantine pest for Egypt, Morocco and Tunisia on the African continent.

The manuscript (MS) conforms to the journal's requirements.

Introduction

It appears that at least parts of the introduction is AI generated according to copyleaks AI detector.

There should be a declaration at the end of the MS in the Acknowledgements as outlined be the journal here:https://journals.plos.org/plosone/s/ethical-publishing-practice#loc-artificial-intelligence-tools-and-technologies

Materials used

line 180: please cite or describe the selective media isolation

line 195: please describe how was the treatment carried outand how long were the larvae exposed to the spores of the different fungi, and when were the paper towels exchanged under the larvae.

Expression of the treatment in l/ha would be also useful for practicing pest managers, so they could compare the dosage to already existing products.

Results

line 267: I think that the supplementary table could be included after this sentence or paragraph, because it contains relevant and valuable information.

lines 279, 289, : "species related interactions" and "Neither species identity" would be more cleared if the authors would include the fungi or fungal words.

I would appreciate images of the fungal colonies the healthy and infected larvae and images of the eaten corn leaves from each treatment to make the MS more visually appealing.

Reviewer #2: Reviewer Comments (PONE-D-25-53276)

Decision: Accept with Minor Revisions

My Specific Comments for Authors

Please address the following points to enhance the clarity and completeness of the manuscript. I believe no additional experiments are required.

1. Abstract and Conclusion Clarity

i. Repetitive Content: The final paragraph of the Abstract, lines 37–43, largely repeats the detailed findings summarized in the preceding Results paragraph, lines 30–37. To improve conciseness, consider synthesizing the final paragraph to focus purely on the validation of the hypothesis and the significance of the findings, rather than re-stating the quantitative results.

ii. Conclusion Sentence Refinement: In the Conclusions section (lines 408-424), the sentence "Their efficacy was strongly influenced by spore concentration and insect developmental stage, with eggs and neonates consistently showing higher susceptibility than later instars" is highly accurate but could be enriched by briefly mentioning the crucial third factor confirmed by your analysis. Table 1 shows that Time After Treatment (DAT) was also a highly significant factor in larval mortality (p < 0.001), with a significant concentration x time interaction (p = 0.010). Please ensure the conclusion reflects all three major factors demonstrated in the data (Dose, Stage, and Time) for complete accuracy.

2. Methods Documentation

i. Fungal Isolate Characterization: The Methods section states that the EPF species were obtained from soil samples collected in the Uluguru Mountains, isolated through selective media, and screened for pathogenicity. For full reproducibility, please briefly clarify how the specific identity of the native isolates (i.e., confirming they were truly Purpureocillium lilacinum and Clonostachys rosea) was achieved. Was this identification based solely on morphology, or was a confirmatory method (such as molecular sequencing/PCR) used? This clarification should be added around lines 178-187.

ii. Rearing Conditions for S. frugiperda: The rearing conditions are well-described (25±2°C, 65±5% relative humidity, 12:12 h L:D) relative humidity, 12:12 h L:D). Please confirm in the Materials and Methods (lines 198-199) that all bioassay test units (larvae and eggs treated with EPFs or control) were maintained under these identical controlled conditions throughout the 9-day observation period, as consistency in environmental parameters is critical for EPF bioassays.

3. Results and Discussion Context

i. LD50 Value Context: The Results section reports that the overall lethal dose (LD50) was estimated at 1.6 x 1012 conidia mL-1. While the manuscript correctly notes that second and third instars required lower concentrations to reach LD50 (Fig 3), the single combined LD50 value of 1.6 x 1012 is significantly higher than most reported values for EPFs against early-stage lepidopterans (e.g., 106 - 108 conidia mL-1 cited in the Discussion for similar assays). Please confirm in the text (around lines 268–269) that this high calculated LD50 (1.6 x 1012) represents the average estimated virulence across all tested instars (L2 to L5), which includes the significantly less susceptible late instars, thus explaining the high magnitude. This clarification will prevent potential misinterpretation of the strain’s overall efficacy.

ii. Justification for Instar Selection: The rationale for excluding the 1st and 6th instars is provided in lines 202-207. Please move this detailed justification to the end of the Collection and rearing of test insect subsection (around line 176), or within the Bioassay subsection, rather than placing it after the description of the mortality measurement protocols (lines 199-202). This slight reorganization will improve the flow of the Methods section.

4. Figure and Table Presentation

i. Figure 1 Legend Clarity: Figure 1 shows the effect of concentration and time on mortality. Table 1 shows that both Concentration and DAT were highly significant. The legend for Fig 1 states: "Different letters above bars indicate significant differences among concentrations (Tukey’s HSD, p < 0.05)". Please ensure the figure itself (or its associated legend/notes) also clearly communicates the significance found across the Days After Treatment (DAT), as this is a fundamental and highly significant finding (p < 0.001).

Overall Assessment

This is a valuable contribution to the literature on biological control in Sub-Saharan Africa. The minor revisions requested aim only to strengthen the presentation and ensure that the powerful statistical conclusions are fully supported by clear text and methodology.

**Do you want your identity to be public for this peer review?** For information about this choice, including consent withdrawal, please see our Privacy Policy

Reviewer #1: No

Reviewer #2: No

---

## [Author Response · Author response to Decision Letter 1]

19 Jan 2026

We have revised the manuscript in response to all comments raised by the academic editor and reviewers. A detailed point-by-point response is provided in the “Response to Reviewers” file, and all changes are highlighted in the tracked-changes version of the manuscript.

---

## [Decision Letter · Decision Letter 1]

27 Jan 2026

Pathogenicity of Purpureocillium lilacinum and Clonostachys rosea against fall armyworm (Spodoptera frugiperda) under laboratory conditions

PONE-D-25-53276R1

Dear Dr. Mussa,

We’re pleased to inform you that your manuscript has been judged scientifically suitable for publication and will be formally accepted for publication once it meets all outstanding technical requirements.

Kind regards,

Yêyinou Laura Estelle Loko

Academic Editor

PLOS One

Additional Editor Comments (optional):

Reviewers' comments:

Reviewer's Responses to Questions

**Comments to the Author**

Reviewer #1: All comments have been addressed

2. Is the manuscript technically sound, and do the data support the conclusions?

Reviewer #1: Yes

3. Has the statistical analysis been performed appropriately and rigorously?

Reviewer #1: I Don't Know

4. Have the authors made all data underlying the findings in their manuscript fully available?

Reviewer #1: Yes

5. Is the manuscript presented in an intelligible fashion and written in standard English?

Reviewer #1: Yes

Reviewer #1: Authors have answered all questions. I recommend the MS to be published, pending the editor's decision.

**Do you want your identity to be public for this peer review?** For information about this choice, including consent withdrawal, please see our Privacy Policy

Reviewer #1: No

---

## [Editor Report · Acceptance letter]

PONE-D-25-53276R1

PLOS One

Dear Dr. Mussa,

I'm pleased to inform you that your manuscript has been deemed suitable for publication in PLOS One. Congratulations! Your manuscript is now being handed over to our production team.

Kind regards,

on behalf of

Dr. Yêyinou Laura Estelle Loko

Academic Editor

PLOS One